# Assessment of Students' Mathematical Skills in Relation to Their Strengths and Weaknesses, at Different Levels of the European Qualifications Framework

**Jacek Stańdo [1]**, **Kamil Kołodziejski [2]** and **Żywilla Fechner [2,*]**

1 Centre of Mathematics and Physics, Lodz University of Technology, 90-924 Łódź, Poland; jacek.stando@p.lodz.pl
2 Institute of Mathematics, Lodz University of Technology, al. Politechniki 8, 93-590 Łódź, Poland
* Correspondence: zywilla.fechner@p.lodz.pl

**Abstract:** Many scientific studies focus on finding the relationship between students' mathematical skills and socio-economic, demographic, and ontogenetic factors. In this publication, we answer the question of how students' mathematical skills are achieved in relation to their strengths and weaknesses, also with regard to the use of mathematics in everyday life. In this article, we examine the relationship between the results of the mock final math exam for eighth grade primary school students/final year high school students and additional math classes, the application of math in everyday life and the greatest difficulties with specific areas of taught material. The study was conducted in Poland on almost ten thousand eighth graders and high school leavers who took part in mock exams online, respectively: eighth-grader exam, and school-leaving maturity exam. The participants of these online exams were asked to respond to a survey that pertained to their math grades, attending additional math classes, their perceived most useful mathematical topics in everyday life and future professional work, and identification of their strengths and weaknesses. In the following paper, the relationships between the answers to the survey questions and the results of the mock online exam are analyzed. The results indicate that there are differences in the area of results of the mock exam and answers about strengths and weakness in mathematical literacy. The analysis of answers about use the mathematical knowledge are different for eight-graders and high-school students. Eight-graders indicate the importance of arithmetic operations while high-school students point out more abstract topics like probability, statistics and geometry. The results of the study are compared to the existing results.

**Keywords:** eighth-grade mock exam; school-leaving mock maturity exam; difficulties in math; everyday-life mathematics

## 1. Introduction

The problem of students' difficulties with mathematics is discussed in many contexts, both from the perspective of the student and the teacher as discussed by Hamukwaya [1], Klymchuk et al. [2], Ramli et al. [3] among others. Researchers are also interested in the relationship of difficulties in acquiring mathematical knowledge in the context of the prospect of potential use of mathematics in everyday life, including professional work for example discussed by Ojose in [4]. Different levels of education are also taken into account: from primary education to higher education, which have been studied by Saeed [5], Udousoro [6] or Hamukwaya [1]. The aim of our research is to examine the relationship between the results of the electronic mock exam in mathematics at the eighth grade level and the final exam with answers to survey questions regarding the use of mathematical skills in the future, and the strengths and weaknesses in mathematics declared by the participants [4,7–24]. The problem of sustainability in mathematics education has been discussed for two decades. Li and Tsay [25] draw attention to the complexity

of the problem of sustainability in teaching mathematics. They show little progress in answering fundamental questions about teaching and learning in the light of sustainable development. They also address the issue of sustainable development in the teacher education process. The problem of sustainability in mathematics in teacher education is addressed by Joutsenlahti and Perkkilä [26]. They discuss the problem of sustainable development both in the general education process and in teaching mathematics. The study concerned the understanding of the a/b symbol at different stages of education. A questionnaire was used during a first-year mathematics teaching course at two Finnish universities. The study indicated pedagogical limitations in teaching the fraction concept. The authors suggested ways to improve the teaching of the fraction concept for sustainable development in math education. Moreno-Pino et al. [27] They discussed the problem of teaching mathematics from the perspective of an academic teacher. The authors emphasize the role of sustainable development in teaching students who, as future teachers, will be responsible for social changes and transformations serving sustainable development. The results presented in our article indicate the need to emphasize the greater use of mathematics in everyday life and to point out the practical aspects of mathematics, which leads to conscious living for sustainable development.

This paper aims to find a connection between the results of mock exams for eighth grade students and a school living maturity exam and answers to the questions from the questionnaire attached to the mock exam. We will analyze the relationship between the total number of points obtained from the mock exam and the following issues: the use of additional mathematics classes in last two years, the final grade in mathematics, strengths and weaknesses in mathematical knowledge declared by the participants, and use of mathematical skills in future work and mathematical issues used in everyday life.

## 2. Literature Review

Phonapichat et al. [7] list five reasons why students have difficulty solving math problems. These are as follows: "(1) Students have difficulties in understanding the keywords appearing in problems, thus cannot interpret them in mathematical sentences. (2) Students are unable to figure out what to assume and what information from the problem is necessary to solving it, (3) Whenever students do not understand the problem, they tend to guess the answer without any thinking process, (4) Students are impatient and do not like to read mathematical problems, and (5) Students do not like to read long problems" (see [7]). The literature of the subject also discusses issues related to students' problems with specific concepts or topics covered in mathematics lessons. In Eisenberg's [8] review article the difficulties associated with the notion of function from both a historical and psychological point of view are described.

Sholeha et al. [9] discuss the results of a qualitative descriptive research conducted on eighth grade 1X students at SMPN (Sekolah Menengah Pertama Negeri) 2 Batang Tuaka. Authors point out several issues in this study, such as small number of research subjects and lack of other data sources e.g., learning test results. Zulfah et al. [10] describe ways to measure solving mathematical problems related to the Pythagorean theorem and solving a system of linear equations on the basis of research conducted on eighth-grade students of junior high school. Puspitarani and Retnawati [11] present their findings concerning problems with tasks related to the Pythagorean theorem based on a study of 8th grade students in SMP (Sekolah Menengah Pertama) 1 Todanan and SMP (Sekolah Menengah Pertama) Muhammadiyah 9 Todanan. Study shows similar results to [1], i.e., students had difficulty understanding and analyzing problems, were not careful when solving problems and too hasty in their rush to solve problems. Students' difficulties with geometry tasks have been discussed by Kuzniak and Rauscher [12], Retnawati et al. [13] and Smith [14]; and those connected with calculus of probability and statistics have been presented by Puspitasari et al. [15] or Garfield and Ahlgren [16]. Puspitasari [15] claims that improving ability to think logically is the key to better handling with statistical and probability problems. However, we cannot compare this result with students point of view,

because research did not contain questory. One of the ways of dealing with difficulties in mathematics is attending additional classes, outside of school. This topic was already taken up in the nineties by Levine and Zimmerman [17]. They discussed the impact of taking additional math classes on future earnings. The likelihood of choosing a specific type of profession traditionally assigned to one gender was also discussed. The results indicated a greater likelihood of better earnings and learning a non-traditional profession for women who were taking math classes. No significant effect of extra math classes was observed in the case of blue-collar workers. In [18] there are published the results of his study on the increase in the participation rate in additional, extracurricular math classes around the world. Differences were detected both between different countries as well as within countries. A dependency was observed between a higher demand for participation in additional mathematics classes and the weakness of the national education system. The research compared the case of Korea and the United States: in the former, private lessons are seen as a threat to the education system and should be subject to legal regulations, while in the USA as support for the education system. A similar issue was raised in the article by Zhang et al. [19]. The study was conducted on Chinese high school students. As in the case of Korea, research showed that parents should choose their children's extracurricular activities appropriately, and the government should issue proper regulations regarding their organization. Discussions about the effectiveness of private lessons at school in terms of future university success were examined by Guill and Boss [20]. The study was conducted in Germany. Most of the respondents asserted that additional classes have an impact on mathematical achievements. However, there was no significant difference between math grades or test scores depending on participation in extracurricular activities.

Regardless of the level of mathematical knowledge, there is a need to apply mathematics in everyday life to a greater or lesser extent. The issue of adequate preparation of students for the everyday use of mathematics has been widely discussed in the literature. Ojose [4] touches upon problems related to the knowledge of mathematics and its use in everyday life. He raises the issue of the essence of mathematics and indicates a list of necessary competences that comprise the knowledge of mathematics. He claims that the school does not provide proper knowledge of mathematics and seeks the reasons for this state of affairs. Putnam [21] describes two lessons taught by Valerie Taft in California. The first one is based on the official textbook and deals with the concept of average. The other one consists in hands-on finding the average based on data prepared by Valerie. Focusing on particular steps and lack of reflection on the calculations cause erroneous determination of averages. Kalchman [22] raises the problem of preparing students for final exams in isolation from the use of mathematics in everyday life. Jansen et al. [23] present conclusions from a study on a population of over 500 adult Dutch citizens on the relationship between fear of mathematics and mathematical skills and their use in everyday life, taking into account the gender of the respondents. Kang et al. [24] address the issue of using Augmented Reality to transmit knowledge in the form of everyday life problems.

### 3. Methodology

In this research we compare two studies carried out in Poland in 2022. They consisted in conducting mock exams for eighth graders and high school leavers using a platform. On the platform, we can monitor: the number of points, the number of entries, the time of solving the task.

The project was organized under the patronage of the Rector of Lodz University of Technology and Stowarzyszenie Nauczycieli Matematyki (The Association of Teachers of Mathematics). Dr. Jacek Stańdo created all the exam tasks, both for the eight-grader exam and the high school leaving exam.

The tasks in both exams were reviewed by specialists from the Regional Examination Board in Lodz.

In Poland, as a result of the reform of the education system in 1999, a system of external examinations was introduced, unified throughout the country. The exams verify

the requirements written in the so-called A curriculum that covers the entire country. Work on exams is supervised by the Central Examination Commission. CKE is responsible for creating examination papers. Examinations are conducted in schools, which are supervised by eight Regional Examination Boards. Pupils take a compulsory exam in mathematics at the end of primary and secondary education. The lists of learning outcomes for primary and secondary schools presented below are consistent with the learning outcomes required by the core curriculum.

The first study was conducted among eighth-grade students between 10th and 20th May 2022. The mock eighth-grader exam included 19 tasks which had been reviewed by specialists from the Regional Examination Board in Lodz. Table 1. presents the assumed learning outcomes which were validated with the use of auto-generated math problems.

**Table 1.** The learning outcomes verified for eighth-grade students.

| Task Number | The Learning Outcome |
|:-----------:|:--------------------:|
| 1 | Analyzes operations with numbers |
| 2 | Finds the value of an angle |
| 3 | Constructs a perpendicular line |
| 4 | Performs operations with numbers |
| 5 | Raises numbers to a power |
| 6 | Analyzes the average |
| 7 | Applies percentages in practical situations |
| 8 | Applies counting methods |
| 9 | Calculates the probability of an event |
| 10 | Constructs figures with axial symmetry and central symmetry |
| 11 | Calculates the surface area of a figure in practical situations |
| 12 | Applies the Pythagorean theorem |
| 13 | Transforms algebraic expressions |
| 14 | Calculates the radius of a circle |
| 15 | Solves a linear inequality |
| 16 | Constructs a parallelogram in the coordinate plane |
| 17 | Validates the described situation |
| 18 | Analyzes the problem situation |
| 19 | Determines the volume of the pyramid |

**Definition 1.** *Let X be a set of objects. A one-dimensional generator task (problem) Z(x) is called a linguistic expression which becomes a task (problem) if an element from the set X is substituted for x.*

**Definition 2.** *Let $X_1 \times X_2 \times \ldots \times X_n$, where $X_1$, $X_2$, ..$X_n$ are sets of objects. An n-dimensional generator problem Z(x) is called a linguistic expression which becomes a problem if an element from the set X is substituted for x.*

**Example 1.** *Eve has $x_1$ kg of strawberries and her brother Adam has $x_2$ kg $x_3$. How many kilograms of strawberries do Eve and Adam have together?*

**Answer 1.** *Eve and Adam have together {if $x_3$ ="="less" " then $2x_1 - x_2$ if $x_3$ =="more" " then $2x_1 + x_2$} kg of strawberries.*

The list of learning outcomes are summarized in Table 1.

*3.1. Description of the Study Group of Eighth Graders*

An invitation to participate in a real-time online mock exam was sent to all primary schools in Poland. School data came from the database of the Education Information System (SIO). 261 primary schools from all voivodeships declared their participation, which constituted about 1.7% of all primary schools in Poland, Figure 1.

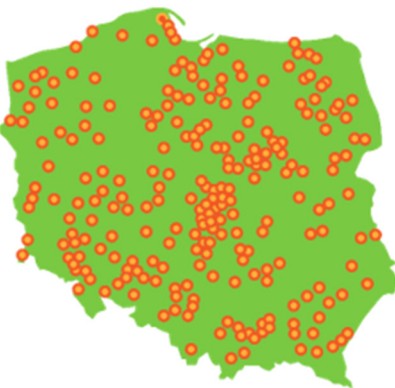

**Figure 1.** Map of schools participating in the project (eight–grade exam).

The participants of the study were 6827 students of the eighth grade of primary schools, who constituted 1.4% of the total population in Poland, Table 2.

**Table 2.** Population of the eighth grader mock exam.

| City or Town of Students' Residence | Data from the Central Examination Board (2022) [28] | | Mock Exam | | % of Population |
|---|---|---|---|---|---|
| | Number | Percentage | Number | Percentage | |
| Town up to 20,000 | 262,576 | 54.23% | 3248 | 47.57% | 1.2% |
| City of 20–100,000 | 97,149 | 23.32% | 1477 | 21.60% | 1.5% |
| City over 100,000 | 118,263 | 24.75% | 2102 | 30.70% | 1.7% |
| Total | 484,174 | | 6827 | | |

The analysis of specific tasks is presented in the Section 4.

### 3.2. The Study Group of High-School Leavers (Graduates)

Between 1–15 April 2022 a nationwide study was conducted that consisted in running a mock high school-leaving math exam online (basic level). The mock exam included 35 tasks. Table 3 presents the assumed learning outcomes.

**Table 3.** The learning outcomes verified for high school leavers.

| Task Number | The Learning Outcome |
|---|---|
| 1 | Determines the equation of the quadratic function |
| 2 | Produces a graph of an exponential function |
| 3 | Applies operations on percentages in practical situations |
| 4 | Determines the equation of a straight line |
| 5 | Applies the abbreviated multiplication formula |
| 6 | Analyzes the arithmetic mean |
| 7 | Defines the domain of the function |
| 8 | Analyzes operations on numbers |
| 9 | Calculates a weighted average |
| 10 | Applies counting methods |
| 11 | Uses progression |
| 12 | Draws a quadratic function |
| 13 | Creates the canonical form of a quadratic function |
| 14 | Finds the value of an angle |
| 15 | Constructs the equation of a line through two points |
| 16 | Calculates the probability of an event |
| 17 | Determines the value of a function based on the graph |
| 18 | Uses properties of logarithms |
| 19 | Constructs a line perpendicular to a given straight line |
| 20 | Analyzes the properties of a prism |

**Table 3.** *Cont.*

| Task Number | The Learning Outcome |
|:---:|:---:|
| 21 | Solves a polynomial equation |
| 22 | Evaluates the trigonometric function for a triangle |
| 23 | Solves a rational equation |
| 24 | Applies theorems about the circumcircle of a triangle |
| 25 | Analyzes problematic situations |
| 26 | Performs operations with numbers |
| 27 | Analyzes the problem using similar figures |
| 28 | Creates figures with central and axial symmetry |
| 29 | Interprets a system of linear equations |
| 30 | Creates an inequality based on the data |
| 31 | Determines the largest and smallest value in an interval |
| 32 | Analyzes the problem situation |
| 33 | Uses arithmetic progression |
| 34 | Constructs a parallelogram with given properties |
| 35 | Determines the surface area and volume of a pyramid |

*3.3. Study Group*

An invitation to participate in a real-time online mock exam was sent to all high schools in Poland. School details were obtained from the Education Information System (SIO) database. 188 high schools (general high schools and technical schools) from all voivodeships declared their participation, Figure 2.

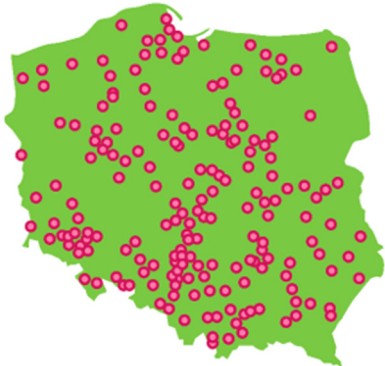

**Figure 2.** Map of schools participating in the project (high-school).

The participants of the study were 3388 students of the final year of high school, who constituted over 1.0% of the total population in Poland, Table 4.

**Table 4.** Population of the high-school leaving mock exam.

| City or Town of Students' Residence | Data from the Central Examination Board (2021) [28] | | Mock Exam | | % of Population |
|:---:|:---:|:---:|:---:|:---:|:---:|
| | **Number** | **Percentage** | **Number** | **Percentage** | |
| Town up to 20,000 | 70,714 | 20.30% | 821 | 24.23% | 1.1% |
| City of 20–100,000 | 126,780 | 36.41% | 1413 | 41.71% | 1.1% |
| City over 100,000 | 150,742 | 43.29% | 1154 | 36.06% | 0.8% |
| Total | 3,480,236 | | 3388 | | |

The aim of this research is to find a connection between the results of mock exams and answers from the questionnaire. Analysis of each answer separately was impossible due to the large sample size. Because of that, we decided to use statistical analysis. Survey responses were grouped in order to check dependencies between survey responses and

exam results. In open questions, grouping based on specific keywords was applied. This method allowed to analyze a large number of possible answers and detect what were the main issues that students struggle with. In order to receive a good representation of the average score and to ignore potential outliers, a median of total exam points was calculated for each group. To check if the differences in medians between groups are statistically significant, Mann–Whitney Wilcoxon tests were performed, which is one of the most popular non-parametric tests for checking differences between non-normally distributed populations. The detailed discussion about the use of non-parametric Mann–Whitney test and using statistical tests in psychology and education can be found in Ferguson and Takane [29].

### 3.3.1. Questions 1–2

Questions 1 and 2 were closed and had 5 possible answers. After removing blank responses (no answer provided to the questionnaire), data was grouped. Each question was considered separately. Then, from the total number of points obtained in the test, medians, averages, and sample sizes were calculated for each of the 5 possible answers and for the entire sample. To show the statistical difference between the means in the groups, the Wilcoxon test was performed for each two pairs of the groups. As a result of the test, the *p* value was returned along with the information whether it is higher than the selected significance level of 0.05.

### 3.3.2. Questions 3–9

The same statistical analyzes were performed for open questions 3 to 9, the only difference being how the responses were grouped. At the beginning, blank responses were excluded, and the remaining responses were prepared by removing Polish characters and changing all capital letters to lowercase. Next, a division into groups: "A", "B", "C" and „D" was carried out according to keywords. A response that contained a word from the list corresponding to a given group was assigned to that group. In the case of conflicts and belonging to several groups at the same time, the most extreme group was selected; that is, for groups ABC, C would be selected, and for BD, D would be selected. Responses that were not classified into any group were placed in the "other" category.
Overall:

- Group A: operations with numbers, practical calculations, including percentages
- Group B: geometry of figures and their properties, calculating circuit length and surface areas (without the Pythagorean theorem)
- Group C: probability and statistics
- Group D: the Pythagorean theorem
- Group E: trigonometry
- Group F: function and differential calculus
- "Other" group: all others

In this paper we investigate six selected questions closely connected with the topic.

## 4. Data Collection and Analysis

The following section offers a detailed analysis of the responses to selected questionnaire items along with a comparison of the eighth graders' and high school leavers' responses. The more detailed description of the questionnaire is in Appendix A.

Question 1. Have you taken extra math classes in the last two years?

- I haven't (group 1)
- I haven't, but I wanted to (group 2)
- I have no opinion (group 3)
- I have, occasionally (group 4)
- I have regularly attended additional math lessons (group 5)

The group names introduced as above are used in Tables 5 and 6 below.

**Table 5.** Summary of the Mann–Whitney Wilcoxon test results—eighth grade students, question 1.

| First Group | Second Group | *p*-Value | Conclusion | Median First Group | Median Second Group |
|---|---|---|---|---|---|
| 1 | all | 0.0203 | reject | 9 | 9 |
| 1 | 3 | 0.0043 | reject | 9 | 8 |
| 1 | 2 | 0.0037 | reject | 9 | 8 |
| 1 | 4 | 0.0000 | reject | 9 | 8 |
| 2 | all | 0.0396 | reject | 8 | 9 |
| 3 | 2 | 0.9488 | accept | 8 | 8 |
| 3 | 4 | 0.6107 | accept | 8 | 8 |
| 3 | all | 0.0412 | reject | 8 | 9 |
| 4 | 2 | 0.6353 | accept | 8 | 8 |
| 4 | all | 0.0025 | reject | 8 | 9 |
| 5 | 1 | 0.5679 | accept | 9 | 9 |
| 5 | all | 0.1396 | accept | 9 | 9 |
| 5 | 3 | 0.0089 | reject | 9 | 8 |
| 5 | 2 | 0.0083 | reject | 9 | 8 |
| 5 | 4 | 0.0003 | reject | 9 | 8 |

**Table 6.** Summary of the Mann–Whitney Wilcoxon test results—high school graduates, question 1.

| First Group | Second Group | *p*-Value | Conclusion | Median First Group | Median Second Group |
|---|---|---|---|---|---|
| 1 | 3 | 0.0023 | reject | 25 | 21 |
| 1 | 2 | 0.0042 | reject | 25 | 21 |
| 1 | 4 | 0.1019 | accept | 25 | 24 |
| 1 | all | 0.1257 | accept | 25 | 24 |
| 2 | all | 0.0180 | reject | 21 | 24 |
| 3 | all | 0.0094 | reject | 21 | 24 |
| 3 | 2 | 0.7876 | accept | 21 | 21 |
| 4 | 3 | 0.0252 | reject | 24 | 21 |
| 4 | 2 | 0.0472 | reject | 24 | 21 |
| 4 | all | 0.5844 | accept | 24 | 24 |
| 5 | 3 | 0.0073 | reject | 24 | 21 |
| 5 | 2 | 0.0136 | reject | 24 | 21 |
| 5 | 4 | 0.4177 | accept | 24 | 24 |
| 5 | 1 | 0.5132 | accept | 24 | 25 |
| 5 | all | 0.6108 | accept | 24 | 24 |

The results for the first question are summarized in Figure 3. The results of Mann–Whitney tests are in Tables 5 and 6.

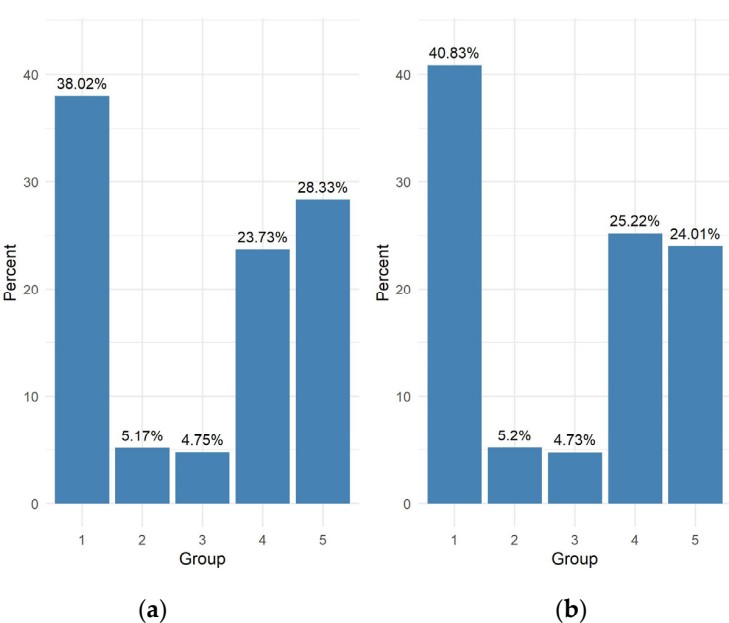

(a)

(b)

**Figure 3.** Summary of responses to question 1 (**a**) eighth graders (**b**) high school graduates.

Summary of question 1

Every fourth student regularly attends additional mathematics classes. On the other hand, about 45% of the students participating in the study do not take any extra classes.

Those eighth graders who did not attend any additional classes and those who attended regularly achieved higher scores than students in other groups. Those students who did not attend additional classes but wanted to and those who attended occasionally achieved lower scores than other students. Those eighth graders who attended extra classes regularly had better scores than those who did not attend but wanted to and better than those who attended occasionally. No statistically significant difference was observed between the median scores of students who regularly attended extra classes, either in relation to the group of students who did not attend additional classes at all, or in relation to the group of all students.

In the group of final year high school students, there were no significant differences between the scores of those who did not attend extra math classes and those who attended regularly (group 5), or those who attended occasionally (group 4) or in relation to all participants (groups 1–5 combined). The study showed that there are no significant differences among high school leavers who regularly attended additional classes in relation to combined groups 1–5.

Question 2. State your math grade on the school-leaving report card (Polish grades are expressed by numbers and their corresponding names: dopuszczający (2)—barely passing, dostateczny (3)—satisfactory, dobry (4)—good, bardzo dobry (5)—very good, celujący (6)—excellent)

- dopuszczający—barely passing (D) (group 1)
- dostateczny—satisfactory (C) (group 2)
- dobry—good (B) (group 3)
- bardzo dobry—very good (A) (group 4)
- celujący—excellent (A plus) (group 5)

The results for the second question are summarized in Figure 4. The results of Mann–Whitney tests are in Tables 7 and 8.

Summary of question 2

**Table 7.** Summary of the Mann–Whitney Wilcoxon test results—eighth grade students, question 2.

| First Group | Second Group | $p$-Value | Conclusion | Median First Group | Median Second Group |
|---|---|---|---|---|---|
| 1 | 2 | 0.0000 | reject | 3 | 6 |
| 1 | all | 0.0000 | reject | 3 | 9 |
| 2 | all | 0.0000 | reject | 6 | 9 |
| 3 | all | 0.0000 | reject | 10 | 9 |
| 3 | 2 | 0.0000 | reject | 10 | 6 |
| 3 | 1 | 0.0000 | reject | 10 | 3 |
| 4 | 3 | 0.0000 | reject | 14 | 10 |
| 4 | all | 0.0000 | reject | 14 | 9 |
| 4 | 2 | 0.0000 | reject | 14 | 6 |
| 4 | 1 | 0.0000 | reject | 14 | 3 |
| 5 | 4 | 0.0000 | reject | 15 | 14 |
| 5 | 3 | 0.0000 | reject | 15 | 10 |
| 5 | all | 0.0000 | reject | 15 | 9 |
| 5 | 2 | 0.0000 | reject | 15 | 6 |
| 5 | 1 | 0.0000 | reject | 15 | 3 |

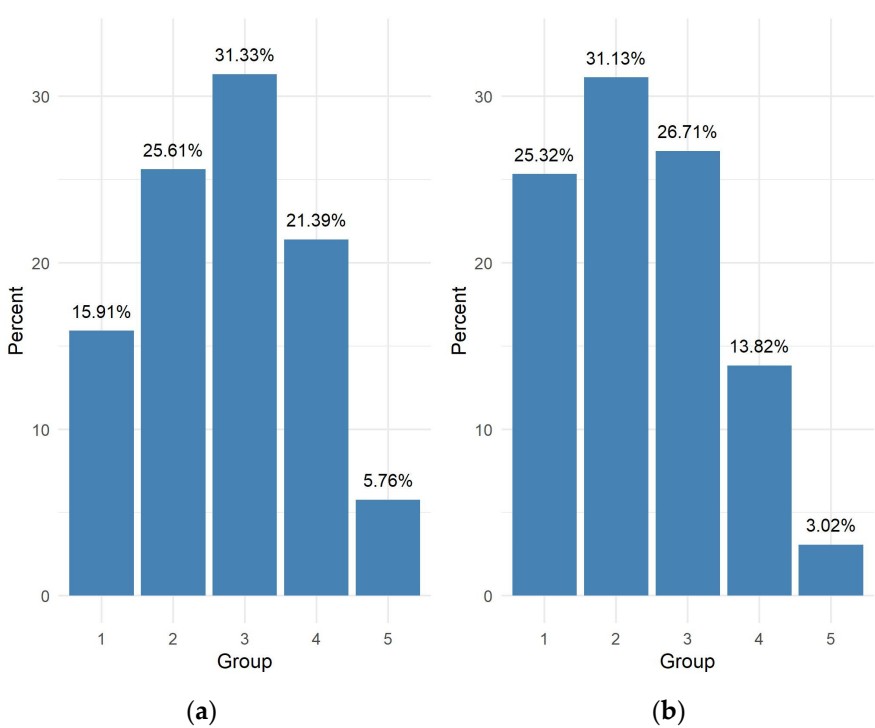

**Figure 4.** Summary of responses to question 2 (**a**) eighth graders (**b**) high school graduates.

**Table 8.** Summary of the Mann–Whitney Wilcoxon test results—high school graduates, question 2.

| First Group | Second Group | *p*-Value | Conclusion | Median First Group | Median Second Group |
| --- | --- | --- | --- | --- | --- |
| 1 | all | 0.0000 | reject | 14 | 24 |
| 2 | 4 | 0.0000 | reject | 21 | 33 |
| 2 | 1 | 0.0000 | reject | 21 | 14 |
| 2 | 5 | 0.0000 | reject | 21 | 34.5 |
| 2 | all | 0.0000 | reject | 21 | 24 |
| 3 | 1 | 0.0000 | reject | 29 | 14 |
| 3 | 2 | 0.0000 | reject | 29 | 21 |
| 3 | all | 0.0000 | reject | 29 | 24 |
| 3 | 4 | 0.0000 | reject | 29 | 33 |
| 3 | 5 | 0.0000 | reject | 29 | 34.5 |
| 4 | 1 | 0.0000 | reject | 33 | 14 |
| 4 | all | 0.0000 | reject | 33 | 24 |
| 4 | 5 | 0.1772 | accept | 33 | 34.5 |
| 5 | 1 | 0.0000 | reject | 34.5 | 14 |
| 5 | all | 0.0000 | reject | 34.5 | 24 |

Eight graders and high school leavers show the same relationship: the better their math grade at the end of school, the higher their test score. Groups of students considered in terms of each of the grades differ significantly (with the exception of "very good" and "excellent" in the case of high school leavers). It should be noted that, in contrast to question 1, homogeneous results are observed here.

Question 3. Which areas of mathematics that you learned in the math class do you consider the most useful in daily life?

The results for the third question are summarized in Figure 5. The results of Mann–Whitney tests are in Tables 9 and 10.

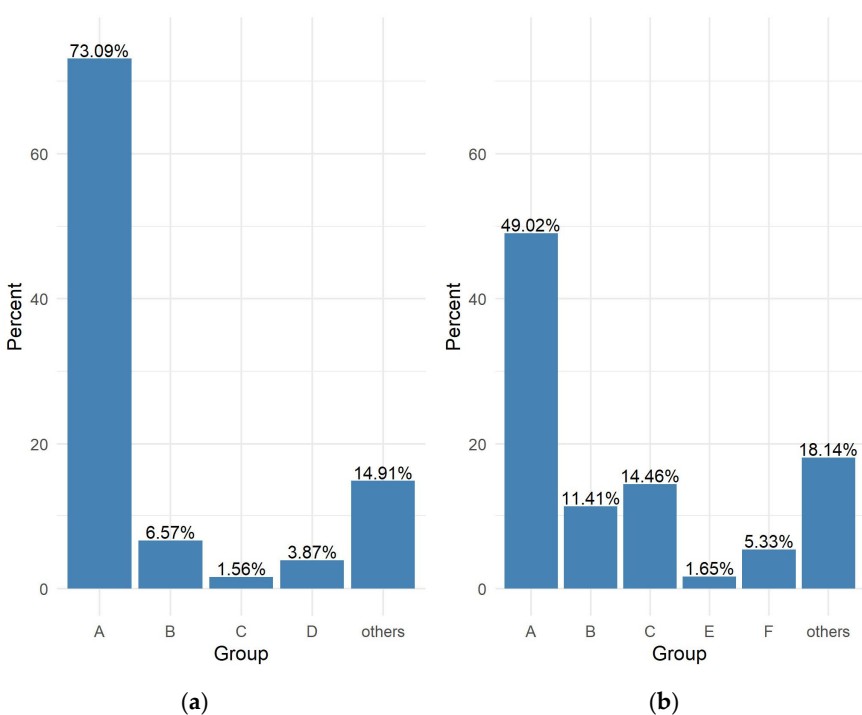

(**a**)                          (**b**)

**Figure 5.** Summary of responses to question 3 (**a**) eighth graders (**b**) high school graduates.

**Table 9.** Summary of the Mann–Whitney Wilcoxon test results—eighth grade students, question 3.

| First Group | Second Group | *p*-Value | Conclusion | Median First Group | Median Second Group |
|:---:|:---:|:---:|:---:|:---:|:---:|
| A | C | 0.0000 | reject | 9 | 13 |
| A | others | 0.0000 | reject | 9 | 7 |
| A | all | 0.8482 | accept | 9 | 9 |
| B | others | 0.0000 | reject | 10 | 7 |
| B | C | 0.0003 | reject | 10 | 13 |
| B | all | 0.0086 | reject | 10 | 9 |
| B | A | 0.0124 | reject | 10 | 9 |
| C | others | 0.0000 | reject | 13 | 7 |
| C | all | 0.0000 | reject | 13 | 9 |
| D | others | 0.0005 | reject | 11 | 7 |
| D | C | 0.0009 | reject | 11 | 13 |
| D | all | 0.1306 | accept | 11 | 9 |
| D | A | 0.1502 | accept | 11 | 9 |
| D | B | 0.7938 | accept | 11 | 10 |
| others | all | 0.0000 | reject | 7 | 9 |

**Table 10.** Summary of the Mann–Whitney Wilcoxon test results—high school graduates, question 3.

| First Group | Second Group | *p*-Value | Conclusion | Median First Group | Median Second Group |
|:---:|:---:|:---:|:---:|:---:|:---:|
| A | C | 0.0000 | reject | 24 | 30 |
| others | C | 0.0000 | reject | 23 | 30 |
| B | others | 0.0001 | reject | 29 | 23 |
| others | F | 0.0002 | reject | 23 | 32 |
| A | B | 0.0003 | reject | 24 | 29 |
| C | all | 0.0003 | reject | 30 | 25 |
| A | F | 0.0003 | reject | 24 | 32 |
| F | all | 0.0046 | reject | 32 | 25 |
| others | all | 0.0098 | reject | 23 | 25 |
| B | all | 0.0131 | reject | 29 | 25 |

**Table 10.** *Cont*.

| First Group | Second Group | *p*-Value | Conclusion | Median First Group | Median Second Group |
|---|---|---|---|---|---|
| A | all | 0.0184 | reject | 24 | 25 |
| others | E | 0.0334 | reject | 23 | 27.5 |
| A | E | 0.0602 | accept | 24 | 27.5 |
| E | all | 0.1751 | accept | 27.5 | 25 |
| B | F | 0.2625 | accept | 29 | 32 |
| A | others | 0.3466 | accept | 24 | 23 |
| B | C | 0.5101 | accept | 29 | 30 |
| C | F | 0.5654 | accept | 30 | 32 |
| B | E | 0.7393 | accept | 29 | 27.5 |
| F | E | 0.8437 | accept | 32 | 27.5 |
| C | E | 0.9899 | accept | 30 | 27.5 |

Summary of question 3

Three in four students in the eighth grade say that operations with numbers and practical calculations, including percentages, are needed in everyday life. However, only one in two high school graduates says so.

The highest scores among eighth graders were achieved by those who considered topics from group C (Probability and Statistics) to be the most necessary in everyday life. The study showed a significant difference in the number of points compared to the group that indicated calculations, including percentages, as most needed in everyday life. Similarly, students indicating probabilistic and statistical issues obtained higher test scores in relation to the other groups, as well as in relation to the group indicating other concepts. Eighth grade students who indicated the topics from group A as the most important in everyday life, obtained lower results than students who indicated group C as the most useful. However, there is no statistically significant difference in the results obtained by eighth graders indicating the most important topics from group A in relation to the group "others".

Among high school leavers, those students who indicated areas from groups B, C and F obtained better results in comparison to students indicating topics other than the ones from groups B, C and F, respectively. However, there are no significant differences between the results achieved by high school graduates indicating topics from groups B, C and F as the most important. As in the case of eighth-graders, the high school graduates who indicated the concepts from group A as the most useful in everyday life achieved significantly lower results in the mock exam than those who indicated the terms from group C as the most useful.

Question 4. What areas of mathematics do you think will be most useful in your future work?

The results for the fourth question are summarized in Figure 6. The results of Mann–Whitney tests are in Tables 11 and 12.

**Table 11.** Summary of the Mann–Whitney Wilcoxon test results—eighth grade students, question 4.

| First Group | Second Group | *p*-Value | Conclusion | Median First Group | Median Second Group |
|---|---|---|---|---|---|
| A | C | 0.0005 | reject | 8 | 13 |
| A | all | 0.1827 | accept | 8 | 8 |
| A | D | 0.3867 | accept | 8 | 9 |
| B | A | 0.0001 | reject | 10 | 8 |
| B | others | 0.0005 | reject | 10 | 8 |
| B | all | 0.0007 | reject | 10 | 8 |
| B | C | 0.0188 | reject | 10 | 13 |
| B | D | 0.3825 | accept | 10 | 9 |

**Table 11.** *Cont*.

| First Group | Second Group | p-Value | Conclusion | Median First Group | Median Second Group |
|---|---|---|---|---|---|
| C | all | 0.0010 | reject | 13 | 8 |
| C | D | 0.0196 | reject | 13 | 9 |
| D | all | 0.5757 | accept | 9 | 8 |
| others | C | 0.0009 | reject | 8 | 13 |
| others | D | 0.4348 | accept | 8 | 9 |
| others | all | 0.4477 | accept | 8 | 8 |
| others | A | 0.8140 | accept | 8 | 8 |

**Table 12.** Summary of the Mann–Whitney Wilcoxon test results—high school graduates, question 4.

| First Group | Second Group | p-Value | Conclusion | Median First Group | Median Second Group |
|---|---|---|---|---|---|
| A | B | 0.0004 | reject | 23 | 27.5 |
| A | C | 0.0000 | reject | 23 | 27.5 |
| A | F | 0.0000 | reject | 23 | 33 |
| A | all | 0.0014 | reject | 23 | 25 |
| B | C | 0.6605 | accept | 27.5 | 27.5 |
| B | F | 0.0004 | reject | 27.5 | 33 |
| B | all | 0.0517 | accept | 27.5 | 25 |
| C | F | 0.0049 | reject | 27.5 | 33 |
| C | all | 0.0085 | reject | 27.5 | 25 |
| E | A | 0.0011 | reject | 31.5 | 23 |
| E | B | 0.1803 | accept | 31.5 | 27.5 |
| E | C | 0.3253 | accept | 31.5 | 27.5 |
| E | F | 0.2785 | accept | 31.5 | 33 |
| E | all | 0.0169 | reject | 31.5 | 25 |
| F | all | 0.0000 | reject | 33 | 25 |
| others | E | 0.0079 | reject | 25 | 31.5 |
| others | A | 0.1415 | accept | 25 | 23 |
| others | B | 0.0135 | reject | 25 | 27.5 |
| others | C | 0.0018 | reject | 25 | 27.5 |
| others | F | 0.0000 | reject | 25 | 33 |
| others | all | 0.2135 | accept | 25 | 25 |

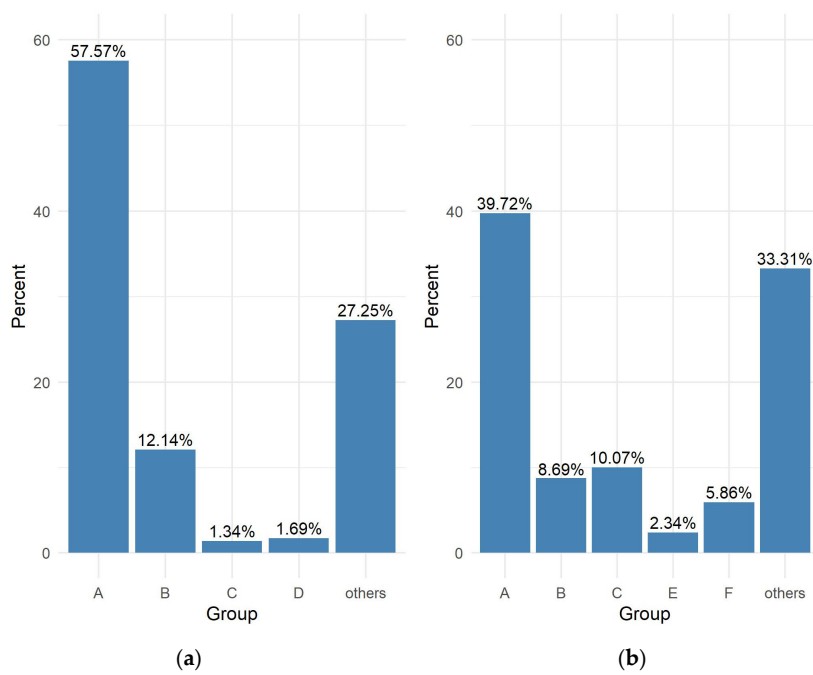

**Figure 6.** Summary of responses to question 4 (**a**) eighth graders (**b**) high school graduates.

Summary of question 4

More than fifty per cent of the eighth-grade students say that operations with numbers and practical calculations, including percentages, will be necessary in their future work. However, this conviction is less firm among high school leavers, who demonstrated increasing awareness of the likely use of probability and statistics in their future work.

Those eighth-grade students who indicated concepts from group C as the most useful, achieved significantly better results in the mock exam compared to other separate groups and the combined group. The eighth graders who indicated the concepts from group B as the most needed ones scored better on the test than the eighth graders who indicated the concepts from group A. Similarly, the results of these study participants were better than those of other respondents who indicated concepts from a group other than B. However, the same students scored lower than eighth graders who indicated concepts from group C.

High school leavers indicating concepts from group C as the most useful in future work obtained better results than those indicating concepts from group A and group "other", while their results were lower than those of high school leavers indicating group F. High school leavers indicating group F as the most useful in future work obtained better results than those indicating groups: A, B, C, "other". High school leavers indicating the concepts from group A as the most needed in future work achieved significantly lower results in relation to participants indicating the issues from groups B, C and F.

Question 5. Which areas of mathematics do you think you would like to understand better, but were difficult for you?

The results for the fifth question are summarized in Figure 7. The results of Mann–Whitney tests are in Tables 13 and 14.

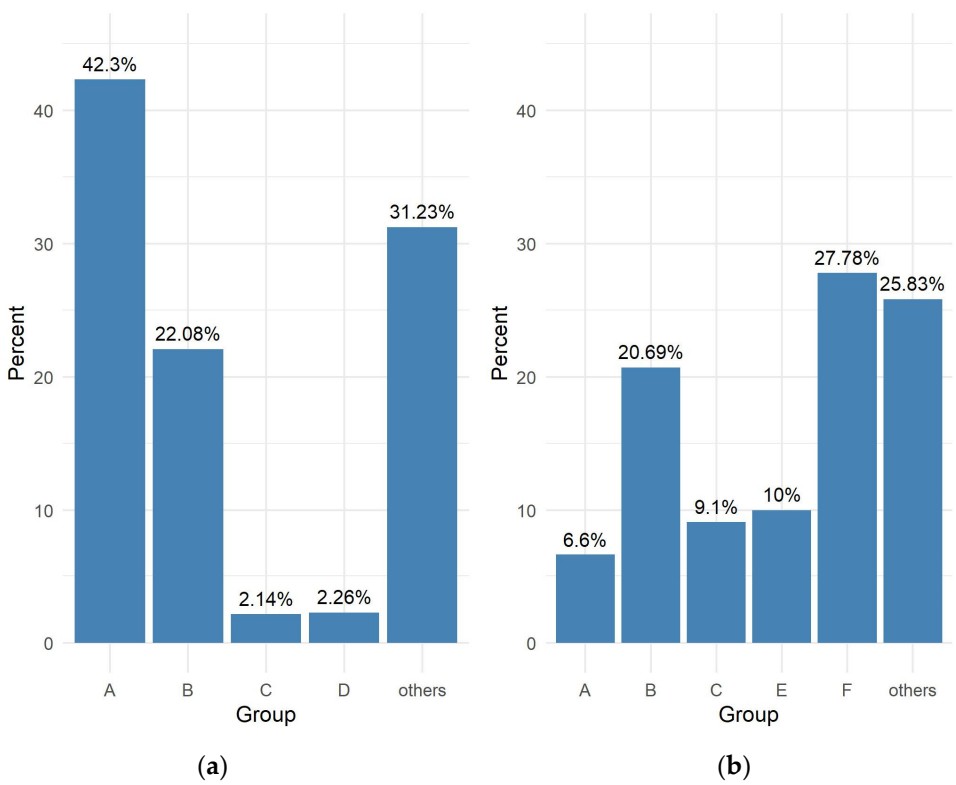

(**a**)  (**b**)

**Figure 7.** Summary of responses to question 5 (**a**) eighth graders (**b**) high school graduates.

**Table 13.** Summary of the Mann–Whitney Wilcoxon test results—eighth grade students, question 5.

| Number of First Group | Number of Second Group | *p*-Value | Conclusion | Median First Group | Median Second Group |
|---|---|---|---|---|---|
| A | others | 0.0841 | accept | 8 | 9 |
| A | B | 0.1357 | accept | 8 | 9 |
| A | all | 0.2385 | accept | 8 | 8.5 |
| B | all | 0.4784 | accept | 9 | 8.5 |
| C | A | 0.0003 | reject | 12 | 8 |
| C | all | 0.0008 | reject | 12 | 8.5 |
| C | B | 0.0020 | reject | 12 | 9 |
| C | others | 0.0026 | reject | 12 | 9 |
| D | C | 0.0000 | reject | 5 | 12 |
| D | B | 0.0001 | reject | 5 | 9 |
| D | others | 0.0002 | reject | 5 | 9 |
| D | all | 0.0003 | reject | 5 | 8.5 |
| D | A | 0.0007 | reject | 5 | 8 |
| others | all | 0.3457 | accept | 9 | 8.5 |
| others | B | 0.8911 | accept | 9 | 9 |

**Table 14.** Summary of the Mann–Whitney Wilcoxon test results—high school graduates, question 5.

| First Group | Second Group | *p*-Value | Conclusion | Median First Group | Median Second Group |
|---|---|---|---|---|---|
| A | E | 0.0000 | reject | 22 | 30 |
| A | all | 0.0011 | reject | 22 | 25 |
| B | C | 0.0001 | reject | 29 | 34 |
| B | others | 0.0000 | reject | 29 | 23 |
| B | F | 0.0000 | reject | 29 | 23 |
| B | A | 0.0000 | reject | 29 | 22 |
| B | E | 0.2979 | accept | 29 | 30 |
| B | all | 0.0033 | reject | 29 | 25 |
| C | others | 0.0000 | reject | 34 | 23 |
| C | F | 0.0000 | reject | 34 | 23 |
| C | A | 0.0000 | reject | 34 | 22 |
| C | E | 0.0019 | reject | 34 | 30 |
| C | all | 0.0000 | reject | 34 | 25 |
| E | all | 0.0004 | reject | 30 | 25 |
| F | A | 0.1830 | accept | 23 | 22 |
| F | E | 0.0000 | reject | 23 | 30 |
| F | all | 0.0004 | reject | 23 | 25 |
| others | F | 0.8759 | accept | 23 | 23 |
| others | A | 0.1951 | accept | 23 | 22 |
| others | E | 0.0000 | reject | 23 | 30 |
| others | all | 0.0029 | reject | 23 | 25 |

Summary of question 5

Eighth grade students most often want to better understand the concepts from group A, and least often the concepts from group C. In the case of high school graduates, the largest number of the respondents want to understand the concepts from group E, and the smallest number the concepts from group A.

Eighth grade students who would like to better understand the Pythagorean Theorem (Group D) obtain low results relative to the other groups, both when considering each group separately and when combined. Very high scores are achieved by eighth graders indicating Group C, who want to better understand probability and statistics. A higher result is observed here both in the case of individual groups and combined groups.

High school graduates who want to understand concepts from groups B and C achieve higher results than those indicating the other groups, while high school graduates indicating

group C achieve significantly higher results than those indicating group B. Poor results are observed in the case of groups A and F.

Question 6. Which math topics do you consider to be your strongest point?

The results for the sixth question are summarized in Figure 8. The results of Mann–Whitney tests are in Tables 15 and 16.

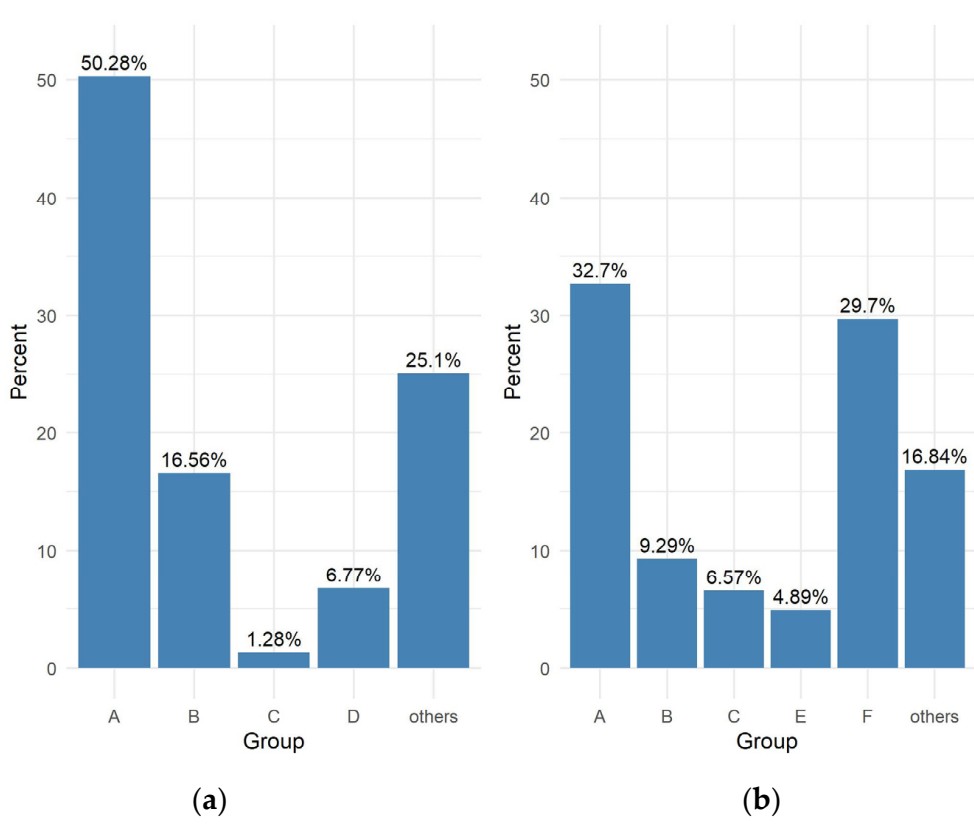

(**a**)                (**b**)

**Figure 8.** Summary of responses to question 6 (**a**) eighth graders (**b**) high school graduates.

**Table 15.** Summary of the Mann–Whitney Wilcoxon test results—eighth grade students, question 6.

| First Group | Second Group | *p*-Value | Conclusion | Median First Group | Median Second Group |
|---|---|---|---|---|---|
| A | D | 0.1186 | accept | 8 | 8 |
| A | C | 0.1571 | accept | 8 | 10 |
| A | all | 0.6822 | accept | 8 | 8 |
| B | others | 0.0000 | reject | 10 | 8 |
| B | D | 0.0000 | reject | 10 | 8 |
| B | all | 0.0000 | reject | 10 | 8 |
| B | A | 0.0001 | reject | 10 | 8 |
| B | C | 0.9021 | accept | 10 | 10 |
| C | D | 0.0371 | reject | 10 | 8 |
| C | all | 0.1757 | accept | 10 | 8 |
| D | all | 0.0675 | accept | 8 | 8 |
| others | all | 0.0470 | reject | 8 | 8 |
| others | C | 0.0697 | accept | 8 | 10 |
| others | A | 0.1318 | accept | 8 | 8 |
| others | D | 0.5558 | accept | 8 | 8 |

**Table 16.** Summary of the Mann–Whitney Wilcoxon test results—high school graduates, question 6.

| First Group | Second Group | *p*-Value | Conclusion | Median First Group | Median Second Group |
|---|---|---|---|---|---|
| A | C | 0.8356 | accept | 21 | 21 |
| A | all | 0.0000 | reject | 21 | 25 |
| B | E | 0.0013 | reject | 29 | 34 |
| B | others | 0.0019 | reject | 29 | 24 |
| B | A | 0.0000 | reject | 29 | 21 |
| B | C | 0.0003 | reject | 29 | 21 |
| B | all | 0.0295 | reject | 29 | 25 |
| C | all | 0.0102 | reject | 21 | 25 |
| E | others | 0.0000 | reject | 34 | 24 |
| E | A | 0.0000 | reject | 34 | 21 |
| E | C | 0.0000 | reject | 34 | 21 |
| E | all | 0.0000 | reject | 34 | 25 |
| F | B | 0.2229 | accept | 30 | 29 |
| F | E | 0.0059 | reject | 30 | 34 |
| F | others | 0.0000 | reject | 30 | 24 |
| F | A | 0.0000 | reject | 30 | 21 |
| F | C | 0.0000 | reject | 30 | 21 |
| F | all | 0.0000 | reject | 30 | 25 |
| others | A | 0.0389 | reject | 24 | 21 |
| others | C | 0.2481 | accept | 24 | 21 |
| others | all | 0.0600 | accept | 24 | 25 |

Summary of question 6

Half of the eighth graders state that their strongest point is operations with numbers and practical calculations, including percentages (group A). In the final year of high school, the number of students in this category significantly decreases.

Eighth grade students who declare geometry (group B) as their strongest point achieve higher results than groups: A, D, "others". The same students achieve results that are not statistically significant in relation to students who indicate group C as their strength.

Among high school graduates, higher results are observed in group B in relation to groups A, C and "others". High school graduates indicating group E score better than any other group. High school graduates indicating group A as their strongest point achieve worse results than those indicating a group other than A. In turn, high school graduates indicating group C as their strongest point achieve significantly worse results than those indicating group B, E or F.

**5. Discussion**

About half of the survey participants declare that they regularly or occasionally attend additional mathematics classes. This indicates that additional mathematics classes are also popular among students in Poland. In line with the assumptions presented by Guill and Boss [20], our study shows that participation in additional classes does not affect the result of the mock exam in mathematics. We do not observe any significant differences in the results of the exam in the case of students attending extra classes when it comes to eighth graders. In light of the research discussed in the introduction, it is worth rethinking whether a central policy for the organization of additional mathematics lessons is necessary. In view of this study, it is not clear whether participation in additional mathematics classes has a positive impact on the results of final exams. What is also worth considering are the reasons for the significant interest in additional lessons among students. The research cited in the introduction shows that in countries with a weaker education system there is more interest in additional mathematics classes than in countries with a better education system. This suggests looking for one of the possible causes in the system that calls for the improvement of mathematical education. Taking a closer look at these issues will make

it possible to try to reduce the scale of additional private classes in Poland for a more balanced development.

The study showed a strong relationship between mock exam scores and math grades. Both in the group of eighth graders and high school graduates, those with higher grades achieved better exam results than those with lower grades. Similar considerations have been applied by Ha et al. [30] for pharmacy students during the pharmaceutical calculations course. About 93% of the students took advantage of the opportunity to take the mock exam. The results of the final exam were significantly higher than the results of the mock exam. This means that the test results were significantly lower than the final grade. In our research, students who performed better on the mock exam received higher math final grades.

The analysis of responses to questions about the use of mathematics in everyday life and in future work shows that at the primary school level, the largest percentage of respondents indicate that topics related to arithmetic calculations, including percentages, are the most useful in the future. However, this percentage decreases in the case of high school graduates who see a greater need, compared to primary school students, to apply topics from the field of probability, statistics, and geometry. Similar observations are presented by Ojose [4], pointing to the need to teach mathematics in a manner consistent with the guidelines of the Program for International Students Assessment (PISA).

The study shows that students of both primary and secondary schools indicate concepts related to arithmetic, practical calculation of percentages, or probability and statistics as the most useful topics applied in everyday life or work. However, the participants did not indicate responses related to logical thinking skills, creating mathematical models to describe the surrounding reality or other, more abstract concepts. For example, the responses did not include topics related to algorithms, widely used for example in programming.

The conducted study demonstrated a difference in the results of the mock exam depending on the areas that students defined as their strengths and weaknesses. It was noticed that students who indicate simple issues as their weaknesses which they would like to overcome, achieve lower results. Eighth graders who pointed to the Pythagorean theorem as the topic they wanted to understand, scored worse in the mock exam. In the case of high school leavers, those students who wanted to better understand issues related to arithmetic calculations, including percentages or functions, achieved low results in the mock exam. One possible explanation is that the difficulty in mastering simpler topics makes it impossible to understand more complex ones.

Analyzing the responses regarding the students' declared strengths in terms of mathematical knowledge, it can be seen that in the case of eighth graders, those declaring geometry or probability and statistics achieve better results than students declaring other categories of concepts. On the other hand, students who declare arithmetic as their strongest point obtain significantly lower results than students who consider other groups of concepts to be their strong points. This might hint at the need to explore in the future the relationship between the way of thinking that is required to analyze geometric or probabilistic problems and raising the general level of mathematical skills. As a continuation of the research, an experiment might be conducted in which two groups of students would be compared: the first one taught in accordance with the curricula set by the educational authorities without additional geometric tasks, and the second one taught fewer typically computational tasks (equivalent to topics from group A) but given extended time devoted to issues in geometry. The results of this study suggest that discussion of issues in geometry can be expected to contribute to the overall improvement of mathematical skills.

## 6. Conclusions

The Recommendation of the Council of the European Union indicates the need to build a European Qualifications Framework (https://eur-lex.europa.eu/legal-content/EN/TXT/PDF/?uri=CELEX:32017H0615(01), accessed on 17 March 2023) For example, an analysis of the answer to the question 4: "What areas of mathematics do you think

will be most useful in your future work"? points out that percentage calculations should be attributed to the second level of the European Qualifications Framework, while the calculus of probabilities, statistics and differential calculus are topics included in at least the sixth level of the European Qualifications Framework for engineers, mathematicians, and economists.

The study has its strengths and weaknesses. One of the strengths of the study is its size. In total, in both groups, almost ten thousand respondents took part. Another asset of the study is the fact that the respondents represented about 1 percent of the group of eighth graders and high school graduates with regard to the size of the town where the student comes from. The electronic form of the test ensured its accessibility regardless of the place of residence. However, this also had a downside: students had unlimited time to solve tasks and complete the survey. The analysis of the time allocated to solve the tasks shows that in some cases it extended to several hours, which suggests that the exam was solved intermittently. As in all surveys, there is no way to verify the accuracy of the answers. In particular, it is impossible to check whether the declared final grade was in line with the facts.

The study was conducted on the population of Polish students and adapted to the Polish education system in the field of mathematics. Taking into account the strengths and weaknesses of the study, it may be the basis for extending the issues to the European Union, where the European Framework of Education applies. Possible future designs of the study should try to limit the weaknesses of the study, such as unlimited response time. It may be considered to extend the IT tools (platform) in a way that enables better verification of the person taking the test.

**Author Contributions:** Conceptualization, J.S. and Ż.F.; methodology, J.S. software, K.K.; validation, J.S., K.K. and Ż.F.; formal analysis, J.S. and Ż.F.; investigation, J.S.; resources, J.S.; data curation, J.S. and K.K.; writing—original draft preparation, J.S. and Ż.F.; writing—review and editing, J.S., K.K. and Ż.F.; visualization K.K.; supervision, J.S.; project administration, J.S. All authors have read and agreed to the published version of the manuscript.

**Funding:** This research received no external funding.

**Institutional Review Board Statement:** Not applicable.

**Informed Consent Statement:** Not applicable.

**Data Availability Statement:** The data was collected during the mock exams described in the paper and is not publicly available.

**Conflicts of Interest:** The authors declare no conflict of interest.

**Appendix A**

Questions in the questionnaire can be divided into two main categories. The first concerns the measurement of the student's knowledge level and the possibility of supplementing knowledge in extracurricular activities (Question one and two), and the second (other questions) concerns the student's subjective views on the strengths and weaknesses of his mathematical skills, the potential use of mathematical knowledge in everyday life, including professional work.

We combine the category concerning the assessment of the student's mathematical knowledge with the results of the mock exam. The final grade in mathematics is an average of the results obtained by the student throughout the year, issued by a given teacher on the basis of partial tests he has checked. The mock exam is centrally structured and the test comparing student skills is the same for all study participants. The results of the mock exam and the final assessment show convergence, so we decided to use the result of the final exam as a criterion for assessing the level of mathematical knowledge of the student. The argument for choosing the results of the trial exam as the criterion for assessing the level of knowledge is its greater objectivity than the final grades. Final scores are determined on the basis of mid-term tests, the level of difficulty of which may vary from school to school.

The second category of questions concerns the student's subjective assessment of the strengths and weaknesses of his mathematical knowledge and the possibility of using mathematics in everyday life and at work. The aim of the study is to find a relationship between the objectively assessed level of knowledge and the subjective assessment of the usefulness of mathematics and difficulties with the acquisition of mathematical knowledge. In general, the higher the level of mathematical knowledge, the more difficult topics are more complicated, and the mathematical tools potentially used in everyday life are more advanced.

Below we present the content of questions for high school and eighth grade.

**Table A1.** Eight grader questionnaire.

| Question | Number of Valid Answers |
| --- | --- |
| Have you taken extra math classes in the last two years? | 4627 |
| State your math grade on the school-leaving report card | 4581 |
| How do you use the Internet to learn math? | 4554 |
| Which areas of mathematics that you learned in the math class do you consider the most useful in daily life? | 3594 |
| Which areas of mathematics that you learned in the math class do you consider the most useless in daily life? | 3516 |
| What areas of mathematics do you think will be most useful in your future work? | 3427 |
| What areas of mathematics do you think will be most useless in your future work? | 3353 |
| Which areas of mathematics do you think you would like to understand better, but were difficult for you? | 3324 |
| Which math topics do you consider to be your strongest point? | 3351 |

**Table A2.** Secondary school questionnaire.

| Question | Number of Valid Answers |
| --- | --- |
| Have you taken extra math classes in the last two years? | 2593 |
| State your math grade on the school-leaving report card | 2621 |
| Which areas of mathematics that you learned in the math class do you consider the most useful in daily life? | 1588 |
| Which areas of mathematics that you learned in the math class do you consider the most useless in daily life? | 1562 |
| What areas of mathematics do you think will be most useful in your future work? | 1460 |
| What areas of mathematics do you think will be most useless in your future work? | 1420 |
| Which areas of mathematics do you think you would like to understand better, but were difficult for you? | 1449 |
| Which math topics do you consider to be your strongest point? | 1440 |
| In what life situation did you need math? | 1388 |
| What is your authority? | 1346 |

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
