# Peer review of "Assessment of Students’ Mathematical Skills in Relation to Their Strengths and Weaknesses, at Different Levels of the European Qualifications Framework"

_sustainability, doi:10.3390/su15118661_

Round 1
Reviewer 1 Report
1. Abstract: Typically an abstract makes reference to the results. Consider adding.
2. Keywords: Eighth grade versus eighth grader? Typically exams reference the grade level.
3. Introduction: "To begin with, we offer a brief review of studies on difficulties in solving math problems" (this sentence is not needed). There are acronyms that are not desc4ibed. On line 41 I think it should be eighth vs eight. Line 51 is an awkward sentence. Line 58, its not necessary to say Lee published his study. Just discuss with a citation. We typically write the "research" versus "he".
4. Materials and Methods: In this article (line 92) or should it say In this research? Line 100. Rather than say Dr. Stando formulated, should just discuss the test and cite Dr. Stando. Line 107 references tasks. Are these test items?
5. Results: Not sure how many students responded in the groups? This is no clear. Was it all 3,388 students?
Conclusions: line 199. Why not say 25% versus 1 of 4?
Overall thoughts. I think this is a good study. I am struggling with the writing. I assume English is a second language. The manuscript needs better use of academic language and English composition. There is a lack of clarity that could be improved if the writing is improved. Consider seeking editorial and composition assistance.
Many of the sentences are awkward. They are not structured properly for an academic paper. This includes word choice. For example, tasks vs. test items. I included other examples within the review. I recommend an editor.
Author Response
Thank you very much for all your remarks and suggestions.
Comments and Suggestions for Authors
Abstract:
- Typically an abstract makes reference to the results. Consider adding. We added more detailed description of the research to the Abstract
Keywords:
- Eighth grade versus eighth grader? Typically exams reference the grade level. It should be: Eighth grade. It is corrected in the manuscript
Introduction:
- "To begin with, we offer a brief review of studies on difficulties in solving math problems" (this sentence is not needed). The sentence has been deleted.
- There are acronyms that are not described. Based on Wikipedia (https://en.wikipedia.org/wiki/List_of_schools_in_Indonesia) we added the full name in Indonesian SMP (Indonesian: Sekolah Menengah Pertama - "First Middle-grade School") Public school (SMPN, SMAN/SMUN and SMKN, with 'N' being Negeri or "State")
- On line 41 I think it should be eighth vs eight. It is corrected in the manuscript.
- Line 51 is an awkward sentence. We are going to use a professional English Editor to correct the manuscript.
- Line 58, it’s not necessary to say Lee published his study. Just discuss with a citation. It is corrected in the manuscript.
- We typically write the "research" versus "he". It is corrected in the manuscript
Materials and Methods:
- In this article (line 92) or should it say In this research? It is corrected in the manuscript
- Line 100. Rather than say Dr. Stando formulated, should just discuss the test and cite Dr. Stando. The word “formulated” has been changed to “created”. It is difficult to cite any particular sources as the tasks were formulated directly on the platform not in a separate article.
- Line 107 references tasks. Are these test items? Yes, this is a sample task.
Results:
- Not sure how many students responded in the groups? This is no clear. Was it all 3,388 students? Two tables with questions from the survey and the number of valid (i.e. non-empty) answers have been added to the appendix.
Conclusions
- line 199. Why not say 25% versus 1 of 4? The meaning is the same. We think that it should be decided by the English service.
Overall thoughts.
I think this is a good study. I am struggling with the writing. I assume English is a second language. The manuscript needs better use of academic language and English composition. There is a lack of clarity that could be improved if the writing is improved. Consider seeking editorial and composition assistance.
Comments on the Quality of English Language
Many of the sentences are awkward. They are not structured properly for an academic paper. This includes word choice. For example, tasks vs. test items. I included other examples within the review. I recommend an editor.
Thank you very much for your comments. We are going to use the Editor Service.
In response to Sustainability editors' suggestions to strengthen the link to the journal's subject matter during the revision, we have added one paragraph at the end of the Introduction section describing the relevance of our research to sustainability
Reviewer 2 Report
This paper focus on the assessment of the mathematical skills of the students concerning to their strengths and weaknesses. The study involved almost ten thousand eighth graders and high school leavers who took part in mock exams online, in Poland. The relationships between the answers given by the students to a survey questionnaire (concerning their math grades, attending additional math classes, their perception of the most useful mathematical topics in everyday life and future professional work, and identification of their strengths and weaknesses) and the results of the mock online exam 23 are analyzed. Considering the number of students involved in this study and the fact that they come from all over Poland, it allows to have a good representativeness of the population under study. Nonetheless, the fact of the exams were online and with no limited time can bias the results. Given this, in order to this paper be accepted to publication, some revision is needed, in accordance with the following points:
1) This study was carried out taking into account the Polish scholar path, which differs from the scholar paths of several other countries. Thus, it must be clear in the paper that, although the obtained results can be considered as inspirations for further analysis in other countries, for the moment they only concern the Polish reality.
2) All over the article, pie charts should be replaced by bar charts, because, somehow, an order can be detected in the groups construction
3) The following observation should also be followed:
Pag 3
- line 113: replace Definition. Let ??×??×…×??, where ??,??,..??_n are sets of objects... " by "Definition. Let ??×??×…×??, where ??,??,..?? are sets of objects... "
None
Author Response
Thank you very much for all your comments and suggestions.
- This study was carried out taking into account the Polish scholar path, which differs from the scholar paths of several other countries. Thus, it must be clear in the paper that, although the obtained results can be considered as inspirations for further analysis in other countries, for the moment they only concern the Polish reality. We added the following text in the last section concerning potential further research: The study was conducted on the population of Polish students and adapted to the Polish education system in the field of mathematics. Taking into account the strengths and weaknesses of the study, it may be the basis for extending the issues to the European Union, where the European Framework of Education applies. Possible future design of the study should try to limit the weaknesses of the study, such as unlimited response time. It may be considered to extend the IT tools (platform) in a way that enables better verification of the person taking the test.
- All over the article, pie charts should be replaced by bar charts, because, somehow, an order can be detected in the groups construction It is already corrected in the manuscript
- The following observation should also be followed:
Pag 3 line 113: replace Definition. Let ??×??×…×??, where ??,??,..??_n are sets of objects... " by "Definition. Let ??×??×…×??, where ??,??,..?? are sets of
- .. " It is already corrected in the manuscript
In response to Sustainability editors' suggestions to strengthen the link to the journal's subject matter during the revision, we have added one paragraph at the end of the Introduction section describing the relevance of our research to sustainability
Reviewer 3 Report
1. I always read the abstract and try to judge it before reading the whole manuscript. In general it is clear and it attracts the reader to continue studying the specific research. The only information which is needed: the country in order to pose the study at a specific framework.
2. The interest concentrates on european qualification framework. For this reason I cannot understand the contribution of the studies in China or Korea, on the specific discussion.
3. About the use of mathematics in everyday life there is a discussion and a different aspect in the case of secondary and higher education
4. The first time there is something about the present study, it is under the section "Materials and Methods" which in my opinion is not the suitable title for the section.
It would be helpful at the end of the previous section to have the purpose of the present manuscript
5. Table 3: We need to understand further the relation of the specific tasks with the Curriculum
6. Indicative examples of the tasks would be useful.
7. There is a confusion for the reader (or for me) between the tasks and the questions.
8. There is a necessity for clear research questions and a section for the presentation of the analyses which is used in each case.
9. At the Tables 5 and 6, I cannot understand what the numbers of the column 1 and 2 mean.
10. I believe that it has to be more clear which are the unexpected results and which is their contribution on the research either for the international community or the specific educational system.
Author Response
Thank you very much for all your suggestions and corrections.
Comments and Suggestions for Authors
- I always read the abstract and try to judge it before reading the whole manuscript. In general it is clear and it attracts the reader to continue studying the specific research. The only information which is needed: the country in order to pose the study at a specific framework.We added the country to the Abstract
- The interest concentrates on european qualification framework. For this reason I cannot understand the contribution of the studies in China or Korea, on the specific discussion. We would like to emphasize that the topic is broadly discussed all over the world. Research concerning Korea and USA show that there is a difference between West and East Culture. In Europe there are also cultural differences between East and West Europe. Potential further studies could take this aspect into account
- About the use of mathematics in everyday life there is a discussion and a different aspect in the case of secondary and higher education. Our aim was to pointed also these aspects in the research.
- The first time there is something about the present study, it is under the section "Materials and Methods" which in my opinion is not the suitable title for the section.
It would be helpful at the end of the previous section to have the purpose of the present manuscript. We added a description also to the Introduction
- Table 3: We need to understand further the relation of the specific tasks with the Curriculum The paragraph concerning curriculum guidelines and their connection to learning outcomes verified during mock exams in Poland has been added to the second section.
- Indicative examples of the tasks would be useful. There are sample tasks in the section Materials and Method
- There is a confusion for the reader (or for me) between the tasks and the questions. The tasks are a part of the test, the questions are part of the questionnaire.
- There is a necessity for clear research questions and a section for the presentation of the analyses which is used in each case.
- At the Tables 5 and 6, I cannot understand what the numbers of the column 1 and 2 mean. The content of columns one and two indicates the answers to question one. For example “1” means “group 1” indicating the answer “I haven’t” for the first question.
- I believe that it has to be more clear which are the unexpected results and which is their contribution on the research either for the international community or the specific educational system. We added one paragraph to the section discussion concerning connection of our research with the international community (lines 526-533)
In response to Sustainability editors' suggestions to strengthen the link to the journal's subject matter during the revision, we have added one paragraph at the end of the Introduction section describing the relevance of our research to sustainability
Reviewer 4 Report
The Abstract provides a concise picture of the research conducted- my suggestion would be authors to expand the third sentence , providing more details and revise the Abstract in the sense of providing not much, but more targeted description on the research. The first sentence of the Introduction could be omitted or revised providing content on the significance of mathematics’ problems – an introduction as such is important so as to provide a smooth transition to the rest of the section. The first paragraph presents important information of literature; however a more critical stance is needed and more elaboration in the writing of this section. At the end of the Introduction a good idea would be to include the aim of the research, and research questions addressed. My suggestion would be authors to revise the Introduction in the sense of a) providing a more critical stance in literature presented, b) use some of this material in the Discussion section, when comparing findings with literature. The Method section needs to be written in a more clear manner ( ie quantitative research, variables, platform used, statistical techniques used and why etc) and justification of these. A description of the educational context could also be helpful ( ie country, curriculum guidelines regarding subject, idiosyncrasy of learning population etc). The content of the questionnaire/quantitative research would be a good idea to be presented in the Methods section, in a short description, highlighting important goals/parameters. In the Research Methodology section a good idea would be authors to include a description of what they present in this section ; for example, the grouping on what basis has been conducted ( ie criteria), why have they selected the Mann Whitney Wilcoxon test etc. An alternative approach in the analysis would be authors to provide description per research item and include the pie charts or other data and include the full questionnaire in the Appendix section- thus the cohesion of the paper would be enhanced. The use of Conclusions caption could be avoided and used only in the last section of the paper. Though the research sample is quite large for the conducted quantitative research, my suggestion would be authors to provide the data and commentary on these in a more cohesive and elaborated manner. In the Discussion section it is important not only to revise the collected data but also comment them in the prism of existing literature in a critical manner. Following that approach both the quality and cohesion of the paper could be enhanced- this research is interesting, however some changes in the presentation/content and structure of the paper are required in my view to enhance cohesion and quality.
The language used is clear and especially in the Introduction section, its use is descriptive and synthetical.
Author Response
Thank you very much for all your suggestions and corrections.
The Abstract provides a concise picture of the research conducted- my suggestion would be authors to expand the third sentence , providing more details and revise the Abstract in the sense of providing not much, but more targeted description on the research We added more detailed description of the research to the Abstract
The first sentence of the Introduction could be omitted or revised providing content on the significance of mathematics’ problems – an introduction as such is important so as to provide a smooth transition to the rest of the section. The first paragraph presents important information of literature; however a more critical stance is needed and more elaboration in the writing of this section. At the end of the Introduction a good idea would be to include the aim of the research, and research questions addressed. My suggestion would be authors to revise the Introduction in the sense of a) providing a more critical stance in literature presented, b) use some of this material in the Discussion section, when comparing findings with literature.
The Introduction and the Discussion section have also been modified taking into account the comments of other reviewers and the editorial office
The Method section needs to be written in a more clear manner ( ie quantitative research, variables, platform used, statistical techniques used and why etc) and justification of these. A description of the educational context could also be helpful ( ie country, curriculum guidelines regarding subject, idiosyncrasy of learning population etc).
The paragraph concerning curriculum guidelines and their connection to learning outcomes verified during mock exams in Poland has been added to the second section.
The content of the questionnaire/quantitative research would be a good idea to be presented in the Methods section, in a short description, highlighting important goals/parameters. In the Research Methodology section a good idea would be for authors to include a description of what they present in this section ; for example, the grouping on what basis has been conducted ( ie criteria), why have they selected the Mann Whitney Wilcoxon test etc. An alternative approach in the analysis would be authors to provide description per research item and include the pie charts or other data and include the full questionnaire in the Appendix section- thus the cohesion of the paper would be enhanced.
The additional tables concerning the questionnaire have been added to the Appendix. The description of the method has been added to section 2.3
The use of Conclusions caption could be avoided and used only in the last section of the paper. We have changed the word “discussion” into “summary of the question” in paragraph 3
Though the research sample is quite large for the conducted quantitative research, my suggestion would be authors to provide the data and commentary on these in a more cohesive and elaborated manner. We added two tables with questions from questionnaires to the Appendix and additional comments to second section.
In the Discussion section it is important not only to revise the collected data but also comment them in the prism of existing literature in a critical manner. We have added references to literature related to similar to ours research.
In response to Sustainability editors' suggestions to strengthen the link to the journal's subject matter during the revision, we have added one paragraph at the end of the Introduction section describing the relevance of our research to sustainability
Round 2
Reviewer 4 Report
The Abstract section could be condensed and present in a clear manner both research and educational context. The first paragraph of the Introduction section could be written in a more critical manner, highlighting important aspects of content provided in relation to the specific research. The reference on specific research papers could be more functional and justified in a Literature Review section and the Introduction section: in the Introduction section it is important to highlight general concepts that underpin the research/educational framework. It is important that authors describe the specific studies in a comprehensible for the reader manner, supporting the readability and quality of their paper. How have the authors compared the studies? This is not clear- a specific research section should be detailed including exact variables for comparison, following a detailed description of the studies. The first paragraph in the Research Methodology section seems a little bit roughly written- my suggestion would be authors to revise it, justifying in a better way their choices and providing references for important research decisions made. What do Questions 1-2 mean? Authors need to find a more functional and understandable for the reader way to present the studies and the analysis they have provided- the cohesion of the paper and coherence in content could be enhanced so as to improve the quality of the paper in this way. A good idea would be authors to include the questionnaire in the last section of the paper- in the Appendix –analytically- but present in corpus a short, clear and understandable description of relative questions. The Discussion section needs to be separate from the Conclusion section- in the Discussion section authors should connect their findings with existing literature in a clear manner, highlighting differences and expressing the added value of their findings. There also has to be reference on questionnaire data and literature references that underpin or differ from research findings. In this section no literature reference has been included. My suggestion would be authors to revise the structure and filter the content of the paper, so as its presentation and content to be more understandable to the reader.
The language used is clear, coherent to academic genre
Author Response
Thank you very much for your remarks and suggestions.
The Abstract section could be condensed and present in a clear manner both research and educational context.
We have changed the abstract. The new part is: The results indicate that there are differences in the area of results of the mock exam and answers about strengths and weakness in mathematical literacy. The analysis of answers about use the mathematical knowledge are different for eight-graders and high-school students. Eight-graders indicate the importance of arithmetic operations while high -school students point out more abstract topics like probability, statistics and geometry. The results of the study are compared to the existing results.
The first paragraph of the Introduction section could be written in a more critical manner, highlighting important aspects of content provided in relation to the specific research.
We have added the following paragraph: The problem of students' difficulties with mathematics is discussed in many contexts, both from the perspective of the student and the teacher as dis-cussed by Hamukwaya [1], Klymcchuk et al. [2], Ramli et al. [3] among others. Researchers are also interested in the relationship of difficulties in acquiring mathematical knowledge in the context of the prospect of potential use of mathematics in everyday life, including professional work for example dis-cussed by Ojose in [4]Different levels of education are also taken into account: from primary education to higher education, which have been studied by Saeed [5], Udousoro [6] or Hamukwaya [1]. The aim of our research is to examine the relationship between the results of the electronic mock exam in mathematics at the eighth grade level and the final exam with answers to survey questions re-garding the use of mathematical skills in the future, and the strengths and weaknesses in mathematics declared by the participants
The reference on specific research papers could be more functional and justified in a Literature Review section and the Introduction section: in the Introduction section it is important to highlight general concepts that underpin the research/educational framework.
We have clarified references to literature on the second page
It is important that authors describe the specific studies in a comprehensible for the reader manner, supporting the readability and quality of their paper. How have the authors compared the studies?
The relationship of our research results with the existing literature is presented in the discussion section
This is not clear- a specific research section should be detailed including exact variables for comparison, following a detailed description of the studies.
We have added the more detailed description to the Appendix
The first paragraph in the Research Methodology section seems a little bit roughly written- my suggestion would be authors to revise it, justifying in a better way their choices and providing references for important research decisions made.
We have revised this paragraph. The present form is: The aim of this research is to find a connection between the results of mock exams and answers from the questionnaire. Analysis of each answer separately was impossible due to the large sample size. Because of that, we decided to use statistical analysis. Survey responses were grouped in order to check dependencies between survey responses and exam results. In open questions, grouping based on specific keywords was applied. This method allowed to analyze a large number of possible answers and detect what were the main issues that students struggle with. In order to receive a good representation of the average score and to ignore potential outliers, a median of total exam points was calculated for each group. To check if the differences in medians between groups are statistically significant, Mann–Whitney Wilcoxon tests were performed, which is one of the most popular non-parametric tests for checking differences between non-normally distributed populations.
What do Questions 1-2 mean? Authors need to find a more functional and understandable for the reader way to present the studies and the analysis they have provided- the cohesion of the paper and coherence in content could be enhanced so as to improve the quality of the paper in this way. A good idea would be authors to include the questionnaire in the last section of the paper- in the Appendix –analytically- but present in corpus a short, clear and understandable description of relative questions. The Discussion section needs to be separate from the Conclusion section- in the Discussion section authors should connect their findings with existing literature in a clear manner, highlighting differences and expressing the added value of their findings. There also has to be reference on questionnaire data and literature references that underpin or differ from research findings. In this section no literature reference has been included.
We have added the more detailed description to the Appendix
My suggestion would be authors to revise the structure and filter the content of the paper, so as its presentation and content to be more understandable to the reader.
Round 3
Reviewer 4 Report
The Abstract could be revised and shortened. In the Introduction section the paragraph beginning with “The aim of the research…” could be transferred in the end of the Introduction since it shuffles the coherence of authors’ rationale. Commentary on the studies examined and used could be used only on the Literature Review section or the Discussion section- in the Introduction it is important to set the conceptual framework . The Introduction section is too long- authors are advised to a)enhance its cohesion by creating robust, long paragraphs in which they present the conceptual framework, b) revise it so as to include parts of it in later sections of the paper. A Literature Review section is missing – it is important authors to enhance the structure of their paper so as to enhance its readability. The format is not well attended. Important sections that it is important to be included are a Literature Review section, Methodology section . The methodology used is not clear- authors need to justify that so as to present in a holistic manner their work. No reference in statistical analysis has been included- the Research Methodology section should be included earlier after a Literature Review section supported by scientific justification ( references, use in context etc). The data presented before the Methodology section puzzles the reader- a firm structure of the paper is required (Introduction, Literature Review, Methodology, Data Collection and Analysis, Discussion, Conclusion). The Data Collection and Analysis section needs to be better organized and presented in a detailed manner so as to enhance the readability of the paper. A better organization of data presentation is required, since otherwise information seem scattered and not comprehensible to the reader. In the Discussion section more reference is needed to prior research and comparison with the data of this work. The Conclusion needs revision to adhere to coherence and cohesion with the rest of the paper
The language used is clear, with good elements of academic genre.
Author Response
Responses to Reviewer #4 (third round)
The Abstract could be revised and shortened.
According to the suggestions of another reviewer, the abstract was extended with a summary of research results, and then rebuilt in the second version of the article
In the Introduction section the paragraph beginning with “The aim of the research…” could be transferred in the end of the Introduction since it shuffles the coherence of authors’ rationale. Commentary on the studies examined and used could be used only on the Literature Review section or the Discussion section- in the Introduction, it is important to set the conceptual framework .
The Introduction section is too long- authors are advised to
a)enhance its cohesion by creating robust, long paragraphs in which they present the conceptual framework,
- b) revise it so as to include parts of it in later sections of the paper.
A Literature Review section is missing – it is important authors to enhance the structure of their paper so as to enhance its readability. The format is not well attended. Important sections that it is important to be included are a Literature Review section, Methodology section.
The introduction has been divided into two parts. References to existing literature have been moved to the literature review section. We want to emphasize that the Sustainability form does not contain such a session, so we included a discussion of the existing literature in the introduction of all versions of the article.
The methodology used is not clear- authors need to justify that so as to present in a holistic manner their work.
We do not understand what is not clear. We added several comments to previous versions.
No reference in statistical analysis has been included- the Research Methodology section should be included earlier after a Literature Review section supported by scientific justification ( references, use in context etc).
We added the reference to the classical textbook by Ferguson and Takane, where the description of all applied statistical methods used in our article is documented.
The data presented before the Methodology section puzzles the reader- a firm structure of the paper is required (Introduction, Literature Review, Methodology, Data Collection and Analysis, Discussion, Conclusion).
The structure of the article has been changed as suggested by the reviewer. We want to reiterate that the Sustainability template does not include a separate Literature Review section. The other reviewers had no comments on the structure of the work.
The Data Collection and Analysis section needs to be better organized and presented in a detailed manner so as to enhance the readability of the paper. A better organization of data presentation is required, since otherwise information seem scattered and not comprehensible to the reader.
This section discusses the questions considered in turn. Each question is described according to the following scheme: the content of the question, bar graphs of answers separately for eighth-graders and high school graduates, the results of the Mann-Whitney tests collected in appropriate tables and a summary of considerations in the form of a verbal description. We don't see how this could be improved. Other reviewers did not comment on the description of this section.
In the Discussion section more reference is needed to prior research and comparison with the data of this work. The Conclusion needs revision to adhere to coherence and cohesion with the rest of the paper
The project was carried out on the basis of the original idea of ​​Dr. Jacek Stańdo and was aimed at linking the result of the test exam (an objective criterion) with the participants' subjective opinions on the usefulness of mathematics in everyday life or subjective difficulties in mathematics. of the problem under study. We are not aware of any references to the conducted research other than those provided in the text of the article.